# Contribution to Improvement of Fatigue Properties of Zr-4 Alloy: Gradient Nanostructured Surface Layer versus Compressive Residual Stress

**DOI:** 10.3390/nano11113125

**Published:** 2021-11-19

**Authors:** Donghui Geng, Qiaoyan Sun, Chao Xin, Lin Xiao

**Affiliations:** 1State Key Laboratory for Mechanical Behavior of Materials, School of Materials Science and Engineering, Xi’an Jiaotong University, Xi’an 710049, China; gengdonghui1130@stu.xjtu.edu.cn (D.G.); (lxiao@xjtu.edu.cn (L.X.); 2XI’AN Rare Metal Materials Institute Co., Ltd., Xi’an 710016, China; xinchao2013@stu.xjtu.edu.cn

**Keywords:** gradient nanostructured surface layer, compressive residual stress, Zr-4 alloy, fatigue properties

## Abstract

The gradient nanostructured (GNS) layer forms beneath the surface of Zr-4 samples by the surface mechanical grinding treatment (SMGT) process, which increases the fatigue strength apparently due to the synergistic effect of the gradient nanostructured layer and compressive residual stress. The SMGTed Zr-4 samples are subjected to annealing to remove residual stress (A-SMGT) and the individual effect of the GNS layer and compressive residual stress can be clarified. The results show that the gradient nanostructure in the surface is stable after annealing at 400 °C for 2 h but residual stress is apparently removed. Both SMGTed and A-SMGTed Zr-4 samples exhibit higher fatigue strength than that of coarse-grained (CG) Zr-4 alloy. The fatigue fracture of Zr-4 alloy indicates that the hard GNS surface layer hinders fatigue cracks from approaching the surface and leads to a lower fatigue striation space than that of CG Zr-4 samples. The offset fatigue strength of 10^6^ cycles is taken for SMRT-ed, A-SMRT-ed, and CG Zr-4 samples and the results indicate clearly that the GNS surface layer is a key factor for the improvement of fatigue strength of the Zr-4 alloy with surface mechanical grinding treatment.

## 1. Introduction

Metals with gradient structures (GS) have attracted great attention in recent years due to a superior combination of strength and ductility [1,2]. A gradient nanostructure is introduced in the surface layer with hundreds of microns by means of surface mechanical attrition treatment (SMAT) or surface mechanical grinding treatment (SMGT), and is characterized by nanoscale to ultra-fine grains and dislocation structures with increasing depth from the surface [3,4,5]. K. Lu’s group has reported that gradient nanograined (GNG) copper film by SMGT exhibits a 10 times higher strength and tensile plasticity comparable to that of coarse grain copper and can sustain a large tensile true strain without cracking when nanograined copper film is confined by coarse-grained substrate. The results indicate that nanograined (NG) metals can be both strong and intrinsically ductile, as long as the strain localization is suppressed. As for the strengthening mechanism of the gradient nanostructure, X.L. Wu has pointed out that strengthening of gradient-structured (GS) metals results from the mechanical incompatibility due to the different strain in GS and coarse-grained (CG) layers upon applied stress, which leads to a two-dimensional stress state and a lateral strain gradient near the elastic–plastic interfaces. Consequently, more dislocations interact and accumulate, resulting in strengthening for the GS metals [6].

Extraordinary strain hardening is another feature of gradient-structured metals, while nanograined and ultrafine-grained metals exhibit little strain hardening, even with strain softening in tensile deformation [7,8]. Strain hardening is not only a strengthening mechanism but also plays an important role in delaying necking of samples with tensile deformation, as well as ductile fracture of metallic materials. The results of X.L Wu’s group exhibit that there is a hardening rate up-turn observed in strain hardening rate versus strain curve of the GS-CG sample, which is different from gradual decrease in the strain hardening rate with strain for the NS and CG samples. By careful TEM observation of dislocation structures, the extraordinary strain hardening rate in the GS-CG sample is attributed to geometrically necessary dislocations (GNDs) by strain gradient, and dislocation accumulation by a multi-axial stress state, according to X.L. Wu’s research [2].

Besides superior combination of strength and ductility in uni-axial tensile tests, the gradient-structured metals also exhibit significant improvement in fatigue strength when subjected to cyclic loads [9,10,11,12]. The researchers conclude that the improved fatigue strength is due to the synergistic effects of compressive residual stress and the gradient nanograined-structured surface layer, which suppresses the initiation of a fatigue crack. As for above factors, which one is predominant depends on specific deformation mechanism of the metal. The researchers found that the GNS surface layer in the 316L samples enhance the fatigue limit more than residual compressive stress [12]. As austenite in 316L stainless is metastable, they further elucidate that the deformation-induced martensite (DIM) in cyclic strain plays an important role in the enhanced fatigue strength in a strain-controlled fatigue regime because formation of DIM during cyclic strain brings about distant second hardening and strain homogeneity, so as to improve the fatigue strength and endurance [9,12]. However, some researchers use an ultrasonic surface rolling process to produce the gradient nanostructured surface layer on Ti-6Al-4V alloy and found a significant improvement in fatigue performance; the authors consider that the main reason is because of the compressive residual stress—although the GNS layer and surface work hardening also contribute to enhancement in the fatigue strength [13].

Zr-4 alloy is widely used as fuel-cladding tubes and grids in commercial nuclear reactors. The microstructure is composed of hexagonal-close-packed α phase and nanoscale second particles, which lead to high strength and good corrosion resistance in the pressurized water and high temperatures in the reactor [14,15,16,17]. Fatigue is one of the failure modes for Zr-4 alloy fuel cladding in nuclear reactors because fuel-cladding tubes are subjected to cyclic stress from the power changes of the start-up and shut-down campaign of the reactor, fluctuation of internal gas pressure inside the cladding tube, and pressurized water temperature outside of the cladding tube, and so on. Additionally, the fatigue properties and mechanism of Zr-4 alloy is investigated and reported by the researchers [18,19]. It requires that Zr-4 alloy maintains good strength as well as good fatigue resistance [20,21].

The gradient nanostructure surface layer from SMGT can simultaneously improve the surface strength as well as the fatigue crack initiation in the fatigue. At the same time, the residual compressive stress generated during the surface gradient nanocrystalline treatment can redistribute the stress in the surface layer during the fatigue process and increase the fatigue life. Therefore, the research on improving the fatigue properties of metals by the surface gradient nanostructure is reported in magnesium, stainless steel, and other materials [22,23,24,25]. As for pure zirconium, the researchers have used ultrasonic shot peening (USSP) to form a gradient nanograined surface (GNS) layer and found that the GNS layer increases fatigue crack initiation resistance and improves the fatigue strength [26,27]. The GNS layer is induced in the surface of Zr-4 alloy by the SMGT process, and the results show that biaxial fatigue strength increases when compared to that of coarse-grained (as-received) samples [28]. The improved fatigue strength results from a complex effect of the compressive residual stress and GNS surface layer.

A gradient nanostructure in the surface is stable when the SMGTed Zr-4 sample is heated to 400 °C according to the results of the reference [29]. Therefore, the residual stress can be removed by heating the SMGTed sample to 400 °C without changing the GNS layer. In the present work, the SMGT process is used to form the GNS surface layer in the surface Zr-4 alloy samples, and tension–compression fatigue tests have been conducted on SMGTed samples (SMGTed Zr-4) and annealed SMGTed samples (A-SMGTed Zr-4), as well as non-treated (as-received) CG Zr-4 alloy samples at room temperature. The aim of the paper is to investigate the comprehensive effect of the gradient nanostructured surface layer and residual compressive stress on the fatigue performance of Zr-4 alloy. In addition, the individual effect of the GNS and compressive stress on fatigue behaviors is separated by removing residual stress without changing the microstructure of the GNS layer. The research results shed light on mechanisms of improvement for fatigue crack initiation and propagation in SMGTed, A-SMGTed, and also supplies more fundamentals for the microstructural design of Zr-4 alloy with higher fatigue resistance.

## 2. Materials and Methods

Zr-4 alloy bars with a 17 mm diameter have been used in the present work. Table 1 shows the chemical composition of Zr-4 alloy.

Surface mechanical grinding treatment (SMGT) is used to induce the gradient structured surface layer beneath the surface of the cylindrical specimens. The details of the SMGT process are found in the references [30]. The GNS specimens are divided into two groups: one group of the samples is SMGTed Zr-4; the other group of the samples is heated to 400 °C for 120 min to remove the compressive residual stress (A-SMGTed Zr-4). Coarse-grained Zr-4 alloy as-received fatigue samples (CG Zr-4) are also measured to compare the change in fatigue properties with different microstructures.

A uni-axial stress-controlled tension–compression fatigue test was carried out on an Instron 1341 electro-hydraulic servo fatigue test machine. The samples for the fatigue tests were dog-bone-shaped specimens with a gauge length of 20 mm and a parallel diameter of 5 mm. The stress amplitude starts from 220 MPa and increases by 20 MPa to the maximum stress amplitude of 320 MPa. The loading frequency in the fatigue tests is 5 Hz, and the stress ratio is R = −1. The axial stress is loaded by a sinusoidal wave. The surface of CG Zr-4 alloy samples was polished with SiC abrasive before the fatigue test, while no polishing was conducted on the SMGTed Zr-4 alloy and A-SMGTed samples.

An FEI Q25 (Philips) SEM was used to observe the fracture morphology of fatigue specimens. A transmission electron microscope (JEM 200CX) was utilized to observe the microstructure of the nanostructure of the surface layer of Zr-4 alloy with different depths for the SMGTed and A-SMGTed samples. TEM sample preparation is shown in Figure 1. The residual stress (macroscopic residual stress) was measured with a Rigaku MSF-3M. The fixed ψ method has been used in this experiment, and six ψ angles (0°, 18.4°, 26.6°, 33.2°, 39.2°, 45°) were selected. In order to measure the residual stress at different depths of the samples, the surface of Zr-4 alloy was peeled by a chemical polishing method.

## 3. Results

### 3.1. Microstructure

The grain size of the as-received sample is about 8 μm, as shown in Figure 2a. The cross-section of the sample after SMGT is shown in Figure 2b. After the SMGT process, a gradient nanostructure with a thickness of about 600 μm forms on the surface of the Zr-4 alloy. Figure 3 shows the microstructure at different depths from the surface of the SMGTed Zr-4 alloy before and after annealing at 400 °C for 2 h. By comparing the TEM images of Figure 3a through e to b through f, it can be observed that there is little change in the grain size after annealing. The submicron grains indicated by the arrows in Figure 3 show that grains did not coarsen in the process of annealing at 400 °C for 2 h. The statistic of the grain size at a distance of 50 to 450 μm from the surface of the SMGTed samples and A-SMGTed samples are shown in Figure 4. The average grain size of the SMGT and A-SMGT Zr-4 alloy is about 161 and 164 nm at 50 μm depth from the surface, respectively, as seen in Figure 4. The grain size increases with the increasing depth distance from the surface for both SMGTed samples and A-SMGTed samples, according to Figure 4.

### 3.2. Compressive Residual Stress

Figure 5 shows the variation of axial residual stress with different depths from the surface of the SMGTed and A-SMGTed Zr-4 alloys. After annealing treatment, the residual compressive stress is apparently eliminated, and the residual stress at different depths from the surface is kept within ±100 MPa. The compressive residual stress of the SMGTed sample is about 300–500 MPa within a 200 μm depth under the surface and it decreases to 100–200 MPa with a depth from 200 to 600 μm. Therefore, it is concluded that the compressive residual stress is eliminated by annealing at 400 °C for 2 h without microstructural change in the GNS surface layer according to Figure 3 and Figure 5.

### 3.3. Fatigue Behaviors

#### 3.3.1. S-N Curve

S-N curves of the SMGTed, A-SMGTed, and CG Zr-4 alloy are exhibited in Figure 6. The results show that the SMGT process increases the fatigue properties when compared to that of the CG Zr-4 alloy. The S-N curves of the A-SMGTed samples are lower than that of the SMGTed samples, but still much higher than that of the CG Zr-4 samples, as seen in Figure 6. This indicates that the increase in fatigue strength in the SMGTed samples is mainly due to the gradient nanostructured surface layer.

For the stress-controlled fatigue tests, the relationship between stress amplitude and fatigue life can be expressed by the Basquin equation:σ_a_ = σ′_f_ (2N_f_)^b^(1)
where σ_a_ is the equivalent stress amplitude, N_f_ is the fatigue life, σ’_f_ is the fatigue strength coefficient, and b is the fatigue strength exponent (Basquin exponent). The Basquin equations are found for the CG Zr-4, SMGTed Zr-4, and A-SMGTed Zr-4 alloys, and obtained by linear fitting of fatigue data in Figure 3, as shown in Table 2.

It can be seen from Table 2 that the fatigue strength coefficient (σ_f_′) and fatigue strength exponent (b) of the SMGTed and A-SMGTed Zr-4 alloy are larger than that of the CG Zr-4 alloy. The fatigue strength coefficient is usually related to the tensile strength [12], while the fatigue strength exponent mainly depends on the strain concentration at the crack initiation stage and the stress gradient at the crack propagation stage [31]. The results show that the tensile strength of both the SMGTed and A-SMGTed Zr-4 alloys are higher than that of the CG Zr-4 alloy [29], which leads to an increase in the fatigue strength coefficient (σ_f_′), as seen in Table 2. The strength exponent (b) is related to cyclic damage during fatigue. The ideal case is that the fatigue strength is equal to the static strength, where b is zero. In practice, the fatigue strength is lower than the static strength; therefore, b is a negative value. For nanograined and ultra-fine-grained metals, it is reported that the fatigue strength coefficient (σ_f_′) and fatigue strength exponent (b) show a trade-off relationship [31]. However, gradient-nanostructured Zr-4 alloys exhibit an increase in both the fatigue strength coefficient and fatigue strength exponent, which is in agreement with results of the 316L stainless steel [12].

#### 3.3.2. Fatigue Fracture

Figure 7 shows the macro-to-micro-scale fracture surface of the CG, SMGTed, and A-SMGTed Zr-4 samples at a stress amplitude of 280 MPa. The fatigue crack initiation (region I), fatigue crack propagation (region II), and final instant rupture (region III) are observed clearly in the macro fracture surface, as seen in Figure 7(a-1,b-1,c-1). The hard GNS surface layer has an effect on the appearance of region II and III; the fatigue crack hardly approaches the surface of the samples due to the high strength of the GNS surface layer in SMGTed and A-SMGTed samples. So, the crack propagation rate decreases because of the GNS layer. With high magnification, it can be observed that a fatigue crack initiates at the surface of the samples, which does not depend on the GNS layer or residual stress state, as Figure 7(a-2,b-2,c-2) shows. As for the crack propagation zones, there are fatigue striations in region II, as Figure 7(a-3,b-3,c-3) shows.

## 4. Discussion

### 4.1. Fatigue Crack Propagation Behavior

In order to compare the fatigue crack growth behavior of the CG, SMGTed, and A-SMGTed Zr-4 alloys, it is necessary to further quantitatively analyze the fatigue crack growth behavior of the Zr-4 alloy.

Fatigue striations are mainly caused by the continuous sharpening and blunting of fatigue cracks due to plastic deformation at the fatigue crack tip [32]. A large number of studies have been conducted on the relationship between fatigue striation spacing and fatigue crack growth rate da/dN [33,34]. Researchers have found that the fatigue striations space is approximately equal to the velocity of fatigue crack propagation [34]. Based on this relationship, the fatigue crack propagation rate can be obtained by measuring the fatigue striations space. The fatigue striations space of Zr-4 samples was measured at certain distances and the following distances were selected: 0.5 mm, 1 mm, 1.5 mm, and 2 mm away from the crack source. An average striation space was measured with more than 50 fatigue striations for each distance, and the average striation space stands for the crack propagation rate in this region. According to the above fatigue striation data, the relationship curves of the fatigue striation space with the fatigue crack length of the CG, SMGTed, and A-SMGTed Zr-4 alloys at the cyclic stress amplitude (280 MPa) are shown in Figure 8. The curves show the fatigue crack propagation rate changes with crack length. A low crack propagation rate is observed when the crack length is less than 1 mm, and the crack propagation rate increases quickly when the crack length is longer than 1 mm. There is little difference in the low crack propagation rate of the CG, SMGTed, and A-SMGTed Zr-4 alloys, while the crack propagation rate of CG Zr-4 alloy is much larger than those of SMGTed and A-SMGTed when the crack length is longer than 1 mm, which is caused by the stress state change and hard GNS surface layer. Moreover, in the fast propagation regime, the GNS surface layer has an effect on the fatigue crack propagation rate instead of compressive residual stress because the fatigue striation space of SMGTed and A-SMGTed Zr-4 samples is very close, as seen in Figure 8.

### 4.2. Enhancement Mechanisms of Fatigue Performance of Zr-4 Alloy with GNS Surface Layer

According to Figure 6, S-N curves of both the SMGTed and A-SMGTed Zr-4 are much higher than that of the CG Zr-4 alloy. The A-SMGTed Zr-4 samples were annealed at 400 °C for 2 h to remove the compressive residual stress, which brings about a little decrease in fatigue performance when compared to SMGTed Zr-4 samples. As for the enhancement mechanism of the fatigue properties of Zr-4 alloy, the following factors work.

The main factor is the nanostructured surface layer, which affects the fatigue properties in two aspects: (1) the crack initiation stage and (2) the crack propagation stage. Firstly, the fatigue crack initiation often occurs on the surface of the sample. After the SMGT process, as reported by our previous results [35], there is a large number of high angle grain boundaries at the depth of 50 μm from the sample surface, which is the main strengthening factor for increased strength of the surface layer. Therefore, the gradient nanostructured surface layer possesses higher strength than the interior part for the SMGTed sample and decreased plastic strain in the fatigue. As for the 316L stainless steel, the gradient nanostructured surface layer obviously inhibits PSB formation on the surface during fatigue [12]. The results indicate that fatigue crack initiation is more difficult in the GNS surface layer than the coarse-grained surface. Moreover, X.L. Wu has pointed out that the GNS surface layer also causes mechanical incompatibility, which leads to a two-dimensional stress-state and lateral strain gradient with geometrically necessary dislocations [6]. As for the Zr-4 alloy, Figure 9 shows the dislocation structure of the SMGTed Zr-4 and A-SMGTed Zr-4 alloy fatigue samples. There are plenty of dislocation structures, such as dislocation tangles, both at 50 and 300 μm depths from the surface. As a result, more dislocation activation and interaction (indicated by arrows in Figure 9) are helpful for plastic deformation, hindering the strain localization during fatigue and increasing the fatigue limit.

During fatigue crack propagation, the results show that the SMGTed Zr-4 samples exhibit a lower fatigue striation space than the CG Zr-4 samples, as Figure 8 shows. This indicates that the gradient nanostructure still plays an active role in crack propagation. According to Figure 7, it can be observed that the final instant fracture zone of the CG Zr-4 alloy is arc-shaped, while the final instant fracture zone of the SMGTed and A-SMGTed Zr-4 alloys is a ring, as the dashed lines indicate in Figure 10. The fracture features indicate that cracks initiate on the surface of the SMGTed Zr-4 alloys and then propagate toward the center, while the GNS surface layer hinders crack propagation to the surface; therefore, a hard surface shell acts as a barrier against crack propagation, thus reducing the crack propagation rate.

Compressive residual stress is also an important factor for improvement of fatigue properties. As for the SMGTed Zr-4 alloy, due to the effect of residual compressive stress on the surface, the equivalent stress is the overlay effect of applied load and residual stress, as shown in the following formula:σ_eq_ = σ_loading_ + σ_residual_(2)
where σ_eq_ is the equivalent stress, σ_loading_ is the applied stress, and σ_residual_ is the residual stress. The cyclic stress changes with time in the tension–compression fatigue process, while the maximum stress decreases because of the residual compressive stress. Therefore, the average stress value σ_𝑚_ decreases:(3)$$\sigma_{m}=\frac{\sigma_{\max}+\sigma_{\min}}{2}$$
where σ_m_ is the average stress, σ_max_ is the maximum cyclic stress, and σ_min_ is the minimum cyclic stress. According to the stress analysis of uniaxial tension and compression fatigue, the maximum normal stress surface is axial. The compressive residual stress reduces the maximum normal stress at the tensile stage. Therefore, decreases in maximum normal stress have an inhibition effect on fatigue crack initiation and reduce the fatigue damage in the tensile stage. Another effect of compressive residual stress causes a closure effect on the fatigue crack tip, which also slows down the crack propagation rate during fatigue.

As for the S-N curves and fatigue date of Figure 6, if the stress amplitude at 10^6^ cycles is taken as the offset fatigue strength (σ_-1_) then the offset fatigue strengths (σ_-1_) of the SMGTed, A-SMGTed, and CG Zr-4 alloys are 272 MPa, 264 MPa, and 233 MPa, respectively. The GNS surface layer and compressive residual stress from the SMGT process raises the fatigue strength by 39 MPa when compared to the CG Zr-4 alloy, and the GNS surface layer in the A-SMGTed sample, due to stress relaxation annealing, raises the fatigue strength by 33 MPa. That is to say, the compressive residual stress increases the fatigue strength by only 8 MPa. In summary, the improvement of the uniaxial tension–compression fatigue performance of the SMGTed Zr-4 alloy is attributed to the synergistic strengthening effect of the surface residual compressive stress and gradient nanostructure, where the gradient nanostructured surface layer contributes to the enhancement of the fatigue performance of the Zr-4 alloy much more than the compressive residual stress.

## 5. Conclusions

(1)The gradient nanostructured surface layer in the Zr-4 samples by the SMGT process is stable after annealing at 400 °C for 2h, while the compressive residual stress is apparently relaxed.(2)The fatigue strengths of the SMGTed and A-SMGTed Zr-4 samples are much higher than that of the CG Zr-4 samples. The fatigue limits of the A-SMGTed Zr-4 samples decrease a little when compared to that of the SMGTed Zr-4 alloy, but are much higher than that of the CG Zr-4 alloy. This means that the GNS layer affects the fatigue limit more than the residual compressive stress for the Zr-4 alloy.(3)The surface gradient nanostructured layer is a key factor for improvement in the tensile–compressive fatigue properties of the SMGTed Zr-4 alloy by delaying crack initiation and decreasing the crack propagation rate.

## Figures and Tables

**Figure 1 nanomaterials-11-03125-f001:**
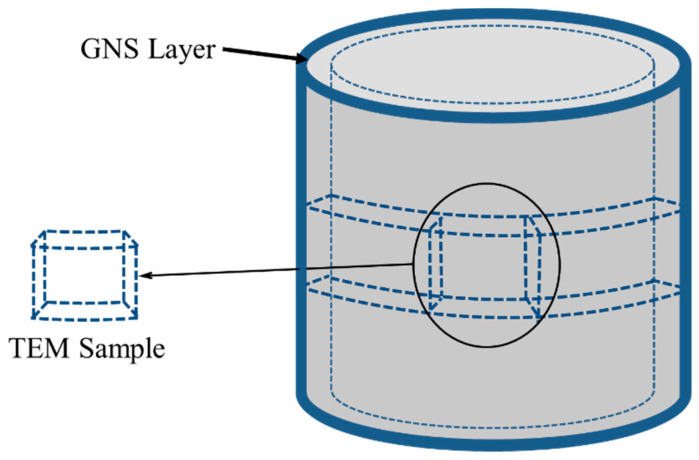
Schematic diagram of TEM sample preparation.

**Figure 2 nanomaterials-11-03125-f002:**
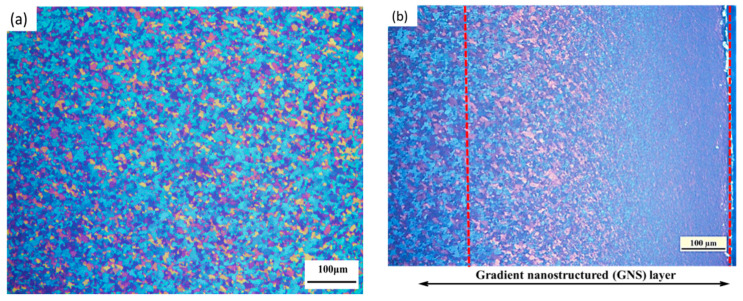
The microstructures of (**a**) as-received samples (CG Zr-4); (**b**) the cross section of Zr-4 alloy after SMGT (SMGTed Zr-4).

**Figure 3 nanomaterials-11-03125-f003:**
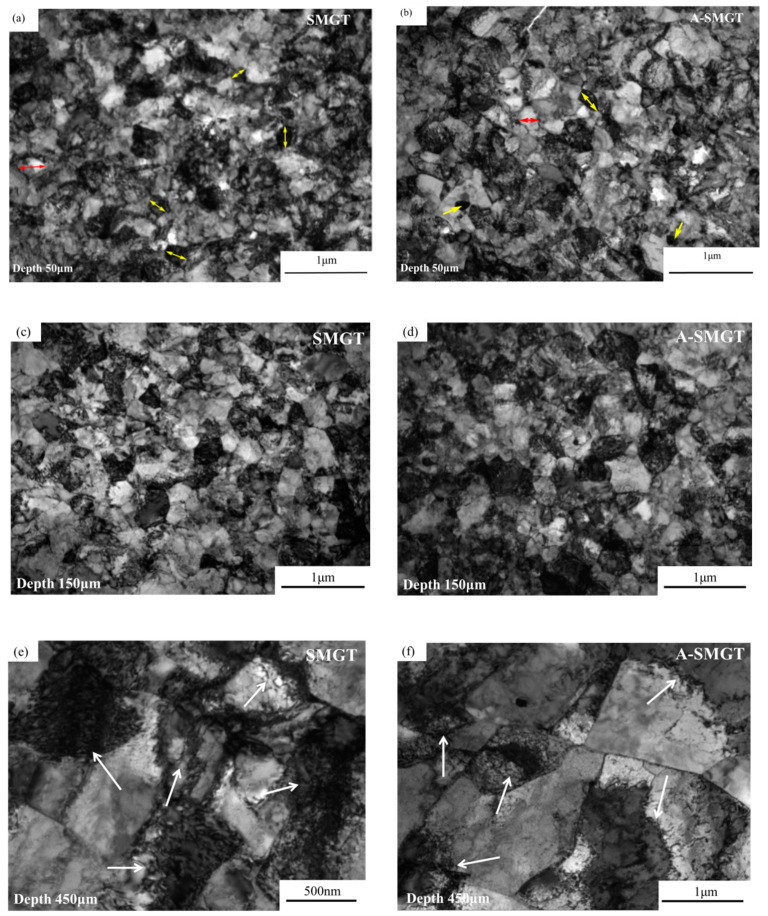
Microstructures of different depths from the surface of SMGTed Zr-4 samples and A-SMGTed Zr-4 samples, SMGTed: (**a**,**c**,**e**) and A-SMGTed (**b**,**d**,**f**). The arrows in (**a**,**b**) indicate the ultra-fined grains.

**Figure 4 nanomaterials-11-03125-f004:**
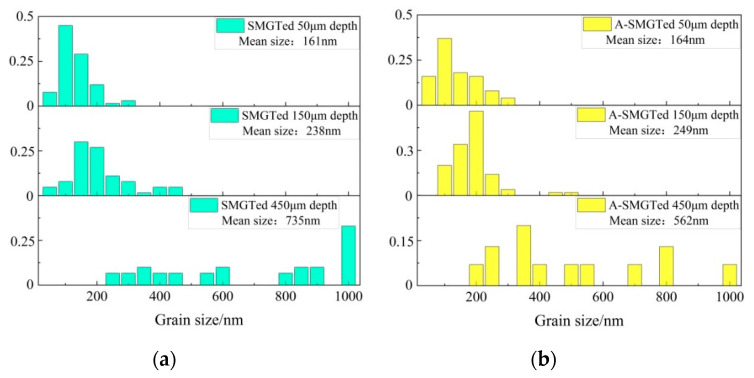
Statistical grain size of (**a**) SMGTed sample. (**b**) A-SMGTed sample. (For Figure 3e,f, grain size is taken as size of dislocation cells, as arrow indicates).

**Figure 5 nanomaterials-11-03125-f005:**
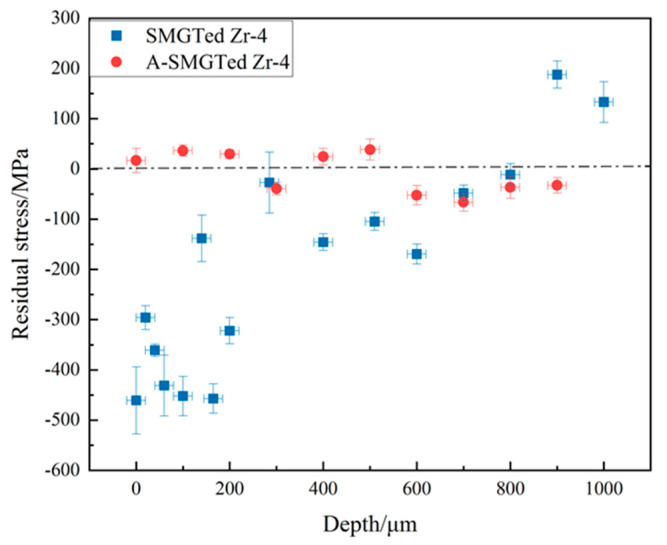
Residual stress distributions with depths of SMGTed and A-SMGTed Zr-4 alloys.

**Figure 6 nanomaterials-11-03125-f006:**
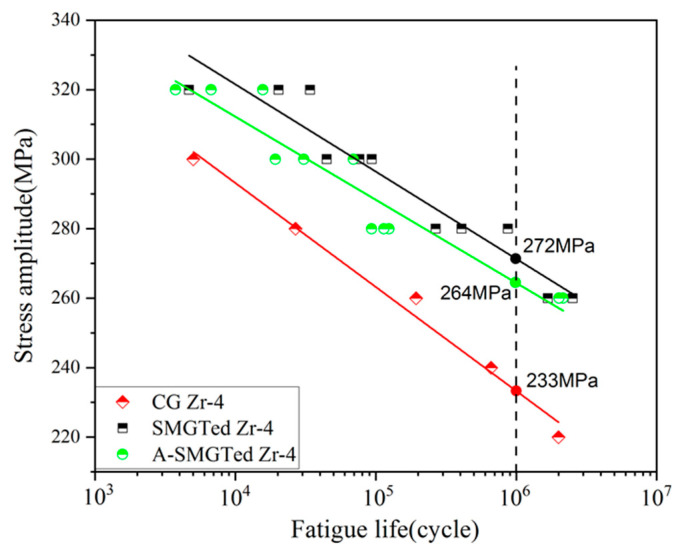
S-N curves of CG, SMGTed, and A-SMGTed Zr-4 samples.

**Figure 7 nanomaterials-11-03125-f007:**
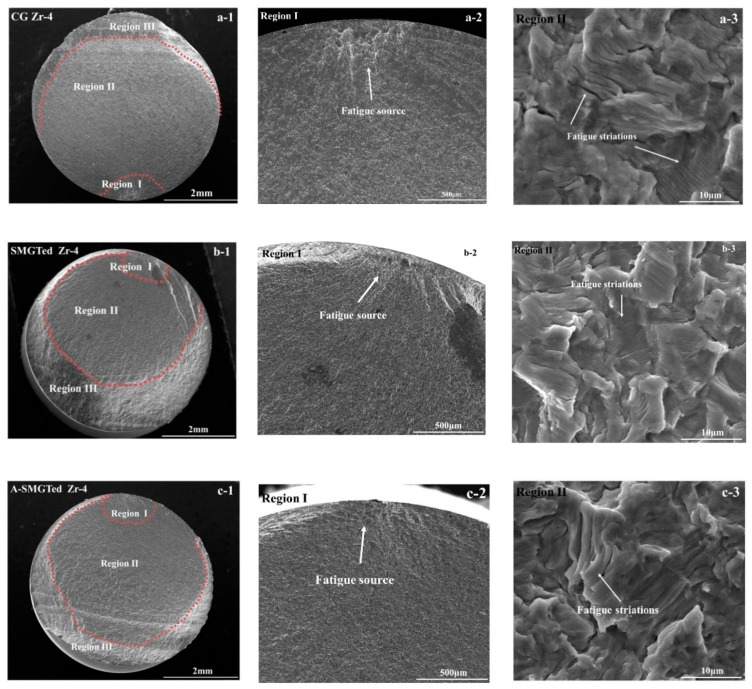
Fatigue fracture of CG, SMGTed, and A-SMGTed Zr-4 samples, (**a-1**,**b-1**,**c-1**) macroscopic fracture morphology of fatigue samples (**a-2**,**b-2**,**c-2**): fatigue crack source; (**a-3**,**b-3**,**c-3**): morphology of fatigue crack propagation region.

**Figure 8 nanomaterials-11-03125-f008:**
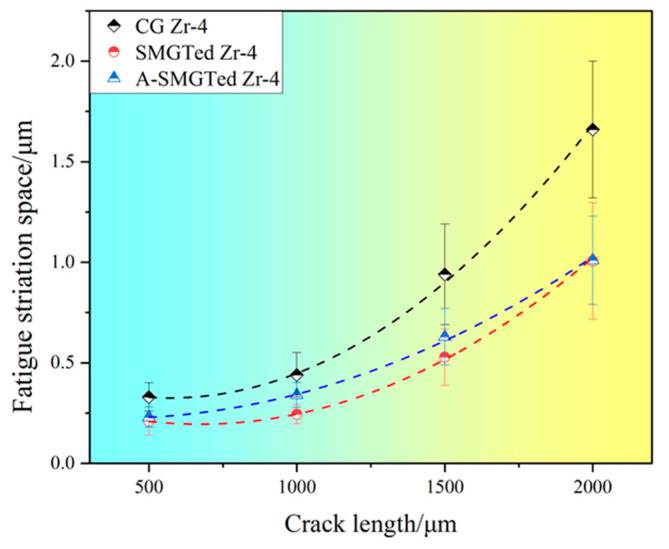
Fatigue striations space versus crack length curves.

**Figure 9 nanomaterials-11-03125-f009:**
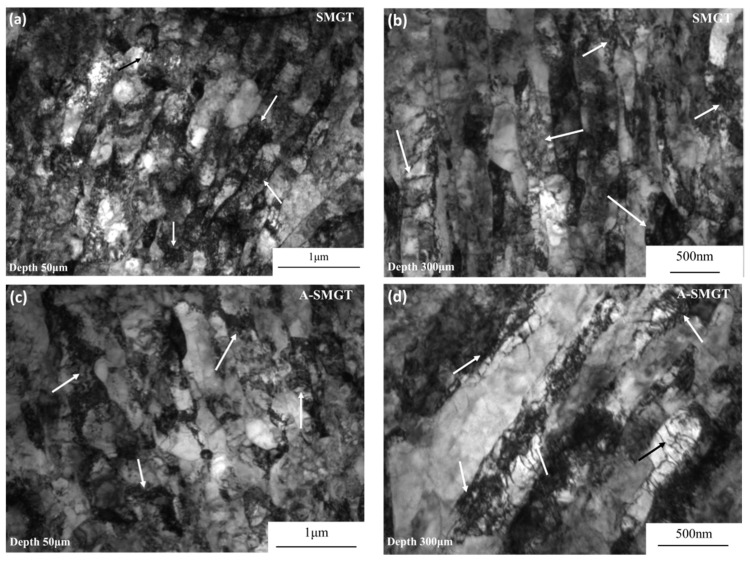
Dislocation structure of different depths from the surface of SMGTed Zr-4 samples and A-SMGTed Zr-4 samples after uniaxial tension–compression fatigue, SMGTed: (**a**) 50 μm and (**b**) 300 μm, and A-SMGTed: (**c**) 50 μm and (**d**) 300 μm. The arrows indicate the dislocation tangles in the gradient nanostructured surface layer.

**Figure 10 nanomaterials-11-03125-f010:**
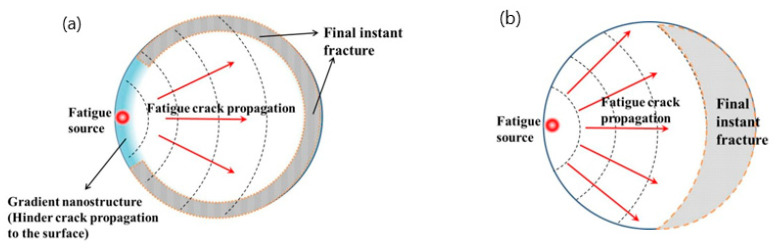
Schematic illustration of fatigue crack initiation, propagation, and final instant fracture, and the effect of SNG surface layer on crack propagation (**a**) SMGTed Zr-4, (**b**) CG Zr-4.

**Table 1 nanomaterials-11-03125-t001:** Chemical composition of Zr-4 alloy/wt.%.

Elements	Sn	Fe	Cr	Hf	Al	Ti	Co	C	H	O	N	Zr
Content	1.4	0.21	0.09	<0.01	0.0015	<0.002	<0.001	0.005	0.001	0.095	0.003	Bal.

**Table 2 nanomaterials-11-03125-t002:** The Basquin equations for CG, SMGTed, and A-SMGTed Zr-4 alloys.

CG	SMGTed	A-SMGTed
σ_𝑎_ = 438(2N_f_)^−0.045^	σ_𝑎_ = 455(2N_f_)^−0.037^	σ_𝑎_ = 455(2N_f_)^−0.040^

## Data Availability

The data presented in this study are available on reasonable request from the corresponding author.

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
