# Peer review of "Contribution to Improvement of Fatigue Properties of Zr-4 Alloy: Gradient Nanostructured Surface Layer versus Compressive Residual Stress"

_nanomaterials, 2021, doi:10.3390/nano11113125_

Round 1

Reviewer 1 Report

The manuscript describes the influence of SMGT induced gradient structured surface layer for improving mechanical properties of  Zr-4 alloy. The paper is written in a comprehensive manner and is worth publishing after several corrections:

.

  1. Fig 1. Fig 1c is marked with f. Please correct.
  2. P 4. “…increases fatigue properties…” – please specify the properties.
  3. P 5. “The previous research…” is not very suitable – please change.
  4. “as seen Table 3 “ change to “as seen in Table 3”.
  5. Improve the English language.
  6. In the microstructure part – if it is it possible, please present a cross-section micrograph together with the presented micrographs for three different depths.
  7. What is the thickness of the obtained surface layer? Please specify.
  8. Is there any phase transformation in the surface layer of the alloy after the SMGT procedure?
  9. Please indicate the role played by annealing by pointing clearly the differences and similarities between the surface treated sample, the annealed one and the non-treated sample.
  10. At the end of “Introduction part” please formulate clearly the aim of the work and its novelty. Specify in more details the considerations about the mechanism of improved performance.

Author Response

Dear Reviewers and Editors,

We really appreciate the comments and suggestions of the editors and reviewers. The title of the manuscript is “Contribution to improvement of fatigue properties of Zr-4 Alloy: Gradient nanostructured surface layer versus compressive residual stress” (Manuscript ID: 1438635). We have carefully considered the comments and have revised the manuscript accordingly. All revisions have been made in the revised manuscript with “track changes”.

The responses to reviewers' comments are listed below point by point. We will be grateful if the manuscript is accepted for publication in Nanomaterials.

Reviewer 2 Report

In my opinion, the manuscript presents results that are new and interesting thanks to a new approach, annealing at moderate temperature to remove the residual stresses. In such a way, the effects of surface grinding on fatigue properties resulted from either decreasing grain size or stress introduced by plastic deformation, or both may be separated. However, I have a great number of remarks which should be taken into account, the additional tests performed, the results, and discussion better elaborated, before the manuscript is allowed to print. They all have been indicated by color in the attached reviewed manuscript and are as follows:

  1. Please present in the introduction for what nuclear applications (what parts) such surface treatment can be applied and how it is related to the failures in this industry.
  2. Please prove that such parts may be subject to fatigue stresses by adding some additional references.
  3. Add also more references demonstrating (ii) how (if there are any) the mechanical surface deformation affects mechanical properties of Zr and its alloys; now, it is only such paper cited and prepared by the authors of the reviewed paper.
  4. In an experiment characterization, there is no description of how the samples for microstructure examinations have been prepared. As concerns Figure 1, there are cross-sections made parallel to the surface at different depths, or are the fragments of a single cross-section made perpendicular to the surface?
  5. The grain size of no-treated alloy, over 80 microns, what is a source of this data?
  6. Regretfully, Fig. 1a and b give no base to accept the average grain size as between 100 and 200 nm. Please prove it, for example by arrows indicating some grains.
  7. 1 does not demonstrate that the density decreases with a depth even if it is reasonable. For such results, the use of TEM is usually applied or an assessment based on the decoration of dislocations. Please prove your assumption or delete these parts of the description.
  8. If the micro or nano-grained layer structure is investigated and the relation between depth and, e.g. grain size, is discussed, the cross-section from the surface to an unchanged zone is necessary to show as a single photo.
  9. Line 109: the surface of Zr-4 alloy was peeled by chemical polishing method; peeled or etched? Please give the name of the reagent.
  10. The authors a few times claim that e.g. “there are a large number of grain boundaries with large and small angles in the gradient nanostructured surface layer, and the grain boundary density also presents a gradient distribution.”. However, I cannot find any evidence on the increasing grain density and the actual indication of low and high angle boundaries.
  11. Then, the authors compare and discuss their results for hard surface layer (and hard alloy) o a very soft copper, e.g. “persistent slip bands easily form in the coarse grained copper during fatigue”. I can agree that such behavior is often observed, but this time, in presence of some precipitates, is less likely and has not been proved. Please give another piece of evidence or references.
  12. In many lines there is a lack of necessary space between, e.g. a value and a unit, and not only: lines 10, 11, 17, 26, 29, 31, 5, 37, 86, 97, 98, 99, 116, 130, 131, 132, 170, 171, 197, 204, 205, 207, 264, 264, 266, 267

Some sentences are difficult to understand:

  1. Line 30: nanoscale-to ultrafined grains and dislocation structures; nanometric to micrometric? What does it mean, dislocation structures?
  2. Line 38: mechanical incompatibility, what does it mean?
  3. Lines 49-50: geometrically necessary dislocations(GNDs); as above.
  4. Lines 50-51; dislocation accumulation by multi-axial stress state; as above.
  5. Line 71: corrosion resistance in performance environment; what does it mean, performance environment?
  6. Lines 92-93: Coarse grained Zr-4 alloy as received fatigue samples (CG Zr-4) are also measured; what does it mean, this sentence?
  7. Lines 223-224: the grain boundary density also presents a gradient distribution. I do not understand, a density is distributed to what?
  8. Line 231-232: more dislocations interaction and accumulation; it hard to understand, what is accumulation? And from what evidence did the authors assume that interaction between dislocations occurs?

There are several grammar errors shown below together with a proposal of modifying the phrases:

  1. Lines 10, 89: in the surface; beneath (under) the surface
  2. Line 18: layer hinder; layer hinders.
  3. Line 18: lead … less fatigue striation space; leads … lower (smaller) striation space.
  4. Line 32: 10 time; 10 times (fold).
  5. Line 36: strengthen; strengthening.
  6. Line 54: fatigue properties is; properties are.
  7. Line 58-59; fatigue properties; fatigue limit.
  8. Line 59: residual compress stress; compressive stress.
  9. Line 63: so as to improved fatigue strength; improving …
  10. Line 69: Zr-4 alloy is widely used as structure materials; … material.
  11. Line 82: for improve fatigue performance; for improving …
  12. Lines 100-101: The surface of CG Zr-4 alloy samples was sanded with SiC abrasive; … was polished with SiC sandpaper.
  13. Line 103: Scanning Electron Microscope was used with FEI Q25; FEI Q25 (Philips) SEM was used.
  14. Line 116: grain size increases as increasing depth; …increases with increasing distance.
  15. Line 131: it decrease; decreases.
  16. Line 160-161: coefficient … as seen Table 3; … as seen in Table 3.
  17. Line 173: crack hardly approach; … approaches.
  18. Line 204: rate increase; rate increases.
  19. Lines 206-207: crack propagation rate of CG Zr-4 alloy is much larger than that of SMGT and A-SMGT; … than those of …
  20. Line 209: crack propagation rage; … rate.
  21. Line 220: main factor is nanostructured surface layers; is … layer.
  22. Line 220-221: from two aspects; in two aspects.
  23. Line 222: there are a large number of grain boundaries; there is …
  24. Line 224: a large number of grain boundaries bring about; … brings about.
  25. Line 235: less fatigue striation space; lower…in this case a space might be assumed as a distance between two striations.
  26. Line 239: fracture features indicate that crack initiation; … crack initiates (starts).
  27. Line 240: surface layer hinder; layer hinders.
  28. Line 242: thus reduce the crack propagation rate; thus reducing …
  29. Lines 259-260: Another effect of compressive residual stress causes closure effect on fatigue crack tip, which also slow down crack propagation rate during fatigue[ … which also slows …
  30. Line 270: gradient nanostructrured surface layer contribute; … nanostructured surface layer contributes.
  31. Lines 274-275: gradient nanostructured surface layer in the Zr-4 samples by SMGT process are stable; … is stable.
  32. Time units (abbreviations) are improperly written: hours instead of h (lines 15, 133, 278), minutes instead of min (line 114).
  33. Besides, the articles are lacking or are improper for several words; check it with, e.g., Grammarly.

Author Response

(The authors gave the same response as above.)

Round 2

Reviewer 2 Report

I have no other comments as all my remarks, in part or fully have been taken into acount

This manuscript is a resubmission of an earlier submission. The following is a list of the peer review reports and author responses from that submission.